# Many-body screening effects in liquid water

Igor Reshetnyak [1,2], Arnaud Lorin[1,2] & Alfredo Pasquarello [1] ✉

The screening arising from many-body excitations is a crucial quantity for describing absorption and inelastic X-ray scattering (IXS) of materials. Similarly, the electron screening plays a critical role in state-of-the-art approaches for determining the fundamental band gap. However, ab initio studies of the screening in liquid water have remained limited. Here, we use a combined analysis based on the Bethe-Salpeter equation and time-dependent density functional theory. We first show that absorption spectra at near-edge energies are insufficient to assess the accuracy by which the screening is described. Next, when the energy range under scrutiny is extended, we instead find that the IXS spectra are highly sensitive and allow for the selection of the optimal theoretical scheme. This leads to good agreement with experiment over a large range of transferred energies and momenta, and enables establishing the elusive fundamental band gap of liquid water at 9.3 eV.

Water is the most common liquid on Earth and plays a fundamental role in physics, chemistry, and biology acting as a medium for numerous reactions and processes. Many efforts have been devoted to measuring and modeling its electronic[1–7] and optical properties[8–11]. The in silico description of liquid water is particularly challenging already at the structural level due to the necessity of accounting for nonlocal van-der-Waals interactions[12,13] and nuclear quantum effects[6,14]. However, high-quality configurations from path-integral molecular dynamics are presently available[15] and can be used to study electronic excited-state properties of liquid water. One of the phenomena that plays a crucial role for the understanding of these properties is the electron screening, i.e., the modification of the potential produced by an electron due to the rearrangement of the other charges. The main contribution to the screening within linear response is provided through the complex dielectric function $\varepsilon(\omega, \mathbf{q}) = \varepsilon_1(\omega, \mathbf{q}) + i\varepsilon_2(\omega, \mathbf{q})$ and can be directly extracted from inelastic X-ray scattering (IXS) experiments. Furthermore, the $\mathbf{q} \to 0$ limit can also be probed in absorption and reflectance measurements.

Numerous IXS experiments have been performed on liquid water over the years with improving resolution both in transferred energy and momentum[9,16–18]. Several semi-empirical models for the dielectric function have been proposed[19–21]. These models assume that the various features of the water spectrum can be traced back to transitions and excitations of the isolated $H_2O$ molecule, which are shifted and broadened in the liquid phase. The shifts and broadenings are unknown parameters and are fitted on experimental data[8,9]. These

models have been successfully applied to both spectroscopic and transport phenomena[22]. Since their predictive power is limited by the heavy reliance on experiments, fully ab initio calculations have been carried out to overcome this drawback. As collective excitations such as excitons and plasmons play a significant role in the screening processes, their treatment requires state-of-the-art approaches such as time-dependent density functional theory (TDDFT) or many-body perturbation theory (MBPT) through the solution of the Bethe-Salpeter equation (BSE). However, results obtained so far are still not fully satisfactory. On the one hand, TDDFT calculations[10,23,24] produced severely red-shifted spectra due to the underestimation of the optical band gap. On the other hand, BSE calculations focused on low-energy excitations[11,25–27] paying less attention to the higher energy ones, which also affect the screening.

As a fundamental property of the system, the screened Coulomb interaction $W$ enters the $GW$ formulation[28], a MBPT approach, which greatly improves the electronic structure of materials with respect to density functional theory (DFT). Even though the occurrence of various $GW$ flavors[29–35] is a source of ambiguity, there are indications that an improved description of the screening, namely through the inclusion of vertex corrections, generally enhances the agreement with experiment[36–40]. The success of the partially self-consistent $GW_0$ in comparison to fully self-consistent $GW$ can also be seen as a consequence of a better description of the screening[41,42]. In the case of liquid water, various flavors of $GW$ have been used for calculating the fundamental band gap[6,7,43–45]. However, the quality of the underlying

[1]Chaire de Simulation à l'Echelle Atomique (CSEA), Ecole Polytechnique Fédérale de Lausanne (EPFL), CH-1015 Lausanne, Switzerland. [2]These authors contributed equally: Igor Reshetnyak, Arnaud Lorin. ✉e-mail: alfredo.pasquarello@epfl.ch

screening has never been assessed resulting in band-gap values ranging from 8.2 to 10.1 eV with an uncertainty considerably larger than that set by experimental references ($9.0 \pm 0.2$ eV[45]).

Here, we present ab initio calculations for the dielectric function of liquid water. We achieve an excellent description of the optical absorption spectrum using various state-of-the-art theoretical approaches. This indicates that an analysis solely based on the optical spectrum is insufficient to discriminate between such approaches, even when they imply different fundamental band gaps. At variance, the IXS spectrum allows us to assess the accuracy of the adopted theoretical schemes and to devise further improvements. Accordingly, the present work also provides reliable values for the energy levels of liquid water, including the fundamental band gap, the exciton binding energy, and the electron affinity.

## Results and discussion

### Absorption

We obtain the optical spectrum of liquid water by solving the BSE, i.e., the state-of-the-art approach, which incorporates many-body effects. Schematically, the equation for the four-point dielectric susceptibility $^4\chi$ reads:

$$^4\chi = {}^4\chi^{IP} + {}^4\chi^{IP}(v+W)^4\chi, \tag{1}$$

where $^4\chi^{IP}$ is the independent-particle four-point susceptibility, $v$ the exchange interaction, and $W$ the screened electron-hole Coulomb interaction. In this framework, neglecting $W$ corresponds to the random phase approximation (RPA). The usual susceptibility $\chi$ is then obtained by contracting $^4\chi$. By construction, $^4\chi^{IP}$ uses single-particle wave functions and energies obtained from $GW$, making it strongly dependent on the adopted scheme. In the following, we will consider as starting point several variants of quasi-particle self-consistent $GW$: partially self-consistent qs$GW_0$ with $W_0$ kept at the DFT level, fully self-consistent qs$GW$ without vertex corrections, and qs$G\tilde{W}$ with vertex corrections included through a TDDFT kernel[38]. We refer the reader to the Methods section for more details.

In Fig. 1, the imaginary parts of the dielectric function $\varepsilon_2$ calculated with various theoretical approaches are compared to experimental spectra obtained from optical reflectance[8] and through Kramers-Kronig transformation of IXS data[17]. The measurements show the main features at the same energies but with different amplitudes. Neglecting the electron-hole interaction corresponds to the RPA, which is found to give an incorrect description. For both qs$GW$ and qs$G\tilde{W}$, this approximation completely fails to describe the first experimental peak at 8.2 eV corresponding to the exciton. Including the electron-hole interaction leads to a qualitative improvement in the dielectric function. The exciton peak at 8.2 eV, the dip at 9 eV, and the peak attributed to transitions to the $\tilde{B}^1 A_1$ state at 9.9 eV[20,46] are then all well reproduced. We emphasize that this agreement is achieved without any alignment between the experimental and theoretical curves. This is at variance with previous $G_0W_0$ calculations[27], in which such an alignment was necessary to compensate for the absence of selfconsistency[11]. The good agreement between theory and experiment is achieved when either qs$GW$ or qs$G\tilde{W}$ are used as starting points in the BSE. This is due to the compensation between the many-body effects in the Green's function $G$ and in the screened electron-hole interaction. In other terms, lower screening leads to an increased fundamental band gap, but concurrently increases the exciton binding energy, a well established effect in semiconductors[47]. In practice, this means that qs$GW$ and qs$G\tilde{W}$ lead to very similar optical spectra, even though they give rather different fundamental band gaps. Therefore, an analysis exclusively based on such spectra cannot assess the accuracy of the adopted schemes in determining the fundamental band gap. This ambiguity also transfers to the exciton binding energy.

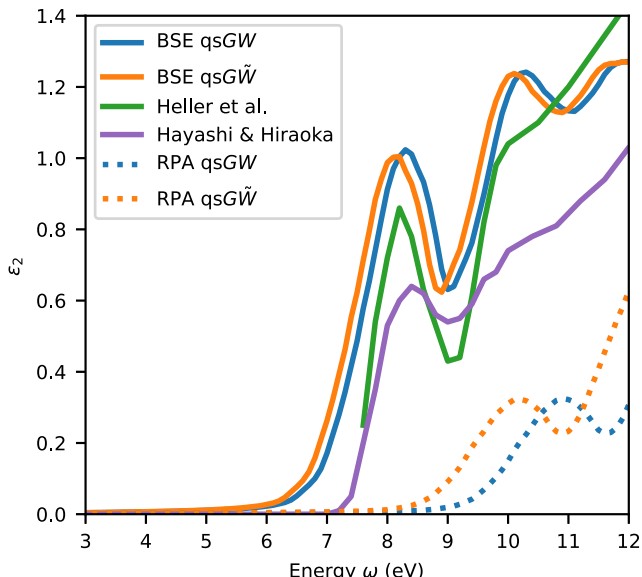

**Fig. 1 | Imaginary part of the dielectric function $\varepsilon_2$ of liquid water.** The calculations are performed at various levels of theory: Bethe-Salpeter equation (BSE) on top of quasiparticle self-consistent $GW$ (qs$GW$) (solid, blue), BSE on top of qs$GW$ with Bootstrap vertex corrections (qs$G\tilde{W}$) (solid, orange), and their respective results in the random phase approximation (RPA) (dotted). Experimental spectra from Hayashi and Hiraoka[17] (purple) and Heller et al.[8] (green) are shown for comparison.

### Inelastic X-ray scattering

The double differential cross section of IXS[48], which measures the energy distribution of X-rays scattered in a solid angle $d\Omega$, is given by:

$$\frac{d^2\sigma}{d\Omega d\omega} = |\mathbf{e}_1 \cdot \mathbf{e}_2^*|^2 \frac{r_0^2 \mathbf{q}^2}{4\pi^2 n}\left(\frac{\omega_2}{\omega_1}\right)\mathrm{Im}\left[\frac{-1}{\varepsilon(\mathbf{q},\omega)}\right], \tag{2}$$

where $\omega_1$ and $\omega_2$ are the frequencies of the incident and scattered X-ray photons, $\mathbf{e}_1$ and $\mathbf{e}_2$ their respective polarizations, $r_0$ is the classical electron radius, $n$ the average electron density, and $\mathbf{q}$ the transferred momentum. Currently, the measurements in the 1–100 eV range are achieved with a resolution of 0.25 eV in energy and of 0.15 Å$^{-1}$ in momentum. Compared to the absorption spectrum, the IXS spectrum is thus measured over a much larger energy region, constituting thereby a formidable reference. The results of IXS experiments are often presented in terms of the loss function $\mathrm{Im}[-\varepsilon^{-1}(\mathbf{q},\omega)]$ or the susceptibility $\chi(\mathbf{q},\omega)$. The two quantities are related through $\varepsilon^{-1}(\mathbf{q},\omega) = 1 + v_c(\mathbf{q})\chi(\mathbf{q},\omega)$, where $v_c$ is the Coulomb potential. To access the large range of energies and momenta measured in IXS experiments, we turn to the TDDFT equation for the two-point susceptibility $^2\chi$:

$$^2\chi = {}^2\chi^{IP} + {}^2\chi^{IP}(v+f_{xc})^2\chi, \tag{3}$$

where $^2\chi^{IP}$ is the independent-particle two-point susceptibility constructed using single-particle wave functions and energies and $f_{xc}$ is the exchange-correlation kernel.

Figure 2 shows calculated loss functions of liquid water in comparison to experimental curves[9]. The loss functions are calculated with qs$GW$ and qs$G\tilde{W}$ energies and wave functions, corresponding to using no $f_{xc}$ and a Bootstrap $f_{xc}$, respectively. The two schemes give noticeably different results in this case: qs$GW$ leads to a strong blue-shift of the spectrum compared to experiment, while qs$G\tilde{W}$ reproduces properly the position of the main peak albeit with an overestimated amplitude. Similar trends are found when these two schemes are applied to crystalline solids[49,50]. Manifestly, the quality of both

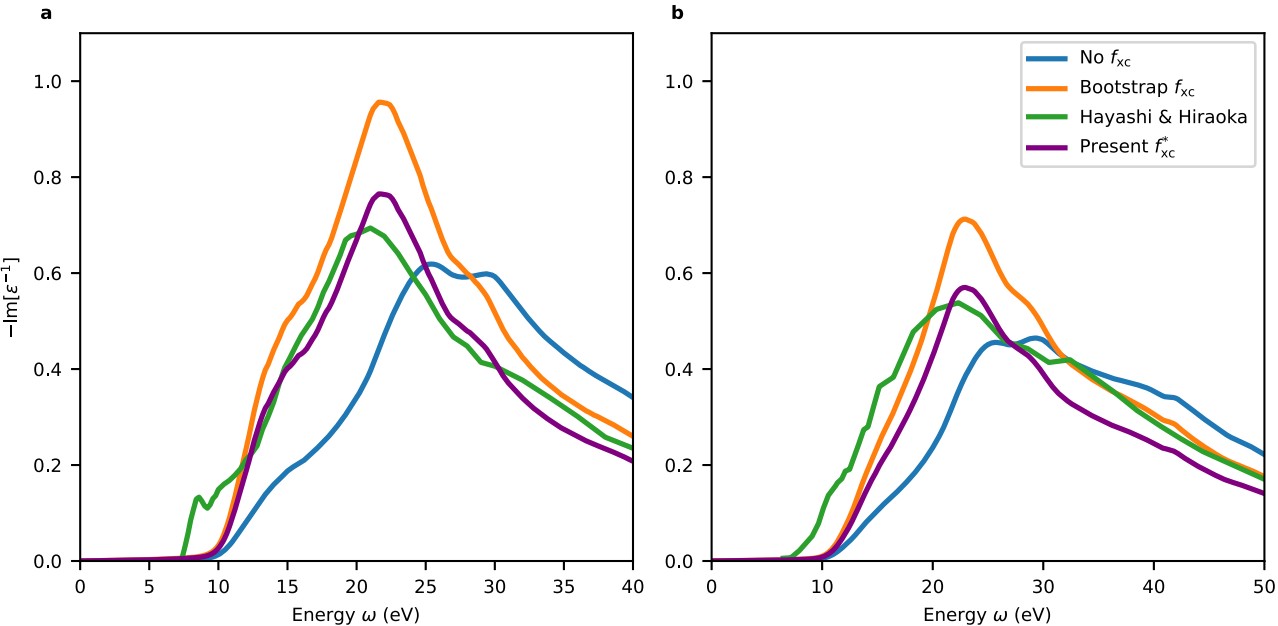

**Fig. 2 | Loss function $-\text{Im}[\varepsilon^{-1}]$ of liquid water at selected transferred momenta.** The transferred momenta correspond to (**a**) $q = 0.32\,\text{Å}^{-1}$ and (**b**) $q = 0.95\,\text{Å}^{-1}$. The results are obtained by solving the time-dependent density functional theory (TDDFT) equation with different exchange-correlation kernels $f_{xc}$: no $f_{xc}$ (blue), the bootstrap $f_{xc}$ (orange), and the modified long-range corrected kernel $f_{xc}^*$ (purple) introduced in this work. Experimental curves from Hayashi and Hiraoka[9] are shown for comparison.

schemes appears unsatisfactory. However, these results indicate a way toward improving the description of the screening through the identification of a suitable $f_{xc}$.

We approach this problem by searching for the optimal kernel within the class defined by:

$$f_{xc}^*(\mathbf{q}, \omega) = \frac{\alpha}{q^2} + \frac{\beta}{\chi(\mathbf{q}, \omega)}, \qquad (4)$$

inspired by the bootstrap kernels[49,51]. Such an expression carries two properties of the exact kernel: it satisfies the asymptotic behavior in the limit $\mathbf{q} \to 0$[50,52–54] and it shows an onset of the dependence on $\omega$ in correspondence of the band gap[55,56]. The parameter $\alpha$ can be related to the dielectric constant[57], while the parameter $\beta$ accounts for a rescaling of the spectral weight to comply with the $f$ sum rule. Here, $\alpha$ and $\beta$ are determined through optimizing the description of the experimental loss function at $q = 0.32\,\text{Å}^{-1}$ given in Fig. 2a. The resulting $f_{xc}^*$ consistently improves the description of the IXS data upon state-of-the-art approaches. Specifically, the intensity of the loss function matches the experimental curve over a broad range of transferred energies and momenta, as shown in Fig. 2 for energies up to 40 eV and two different values of $q$. Such a global improvement suggests that the functional form of $f_{xc}^*$ is able to describe the overall dependence of the loss function. In particular, we remark that the good agreement of the intensities around the peak is important to properly capture the screening effects[58].

**Absence of plasmon**
In Fig. 3a, we show $-\text{Im}[\chi(\omega, \mathbf{q})]$ calculated in TDDFT with and without the $f_{xc}^*$ kernel in comparison with its experimental counterpart[16]. Both theoretical results show a qualitative agreement with experiment over a large range of transferred energies and momenta and, in particular, reproduce the distinctive feature at $q = 2\,\text{Å}^{-1}$. More remarkably, the inclusion of $f_{xc}^*$ brings the theoretical result almost in coincidence with experiment as far as both the peak energy (23–25 eV) and its amplitude (1.5–1.7 as $\text{Å}^{-3}$) are concerned. The main feature has repeatedly been

attributed to a plasmon in the literature[8,59], whereas our calculations do not support this interpretation. Indeed, the real part of the dielectric function $\varepsilon_1(\mathbf{q}, \omega)$ is not only non-vanishing but does not even approach the zero line between 22 to 25 eV for the relevant transferred momenta (cf. Fig. 3b). Therefore, no plasmon wave can propagate in the medium. This is consistent with an analysis based on the Fano criteria[60] and explains why the corresponding excitation dies out after a few oscillations in real-time dynamics[16].

**Energy levels**
Having verified the quality of the screening, we construct an improved screened potential $W^* = \varepsilon^{-1} v_c$ to be inserted in a qs$GW$ calculation of the fundamental band-gap. In Table 1, we compare the band gaps obtained using various forms of the screened potential. When the screening is kept at the DFT level ($W_0$), we obtain a band gap of 8.7 eV. Next, the self-consistent schemes with vertex corrections (qs$G\tilde{W}$) and without vertex corrections (qs$GW$) give higher band-gap values, reaching 9.1 and 9.8 eV, respectively. All these approaches are state-of-the-art and it is thus not possible to narrow the spanned interval of 1.1 eV on the basis of theoretical considerations alone. The present qs$GW^*$ scheme overcomes this limitation as it is designed to account for the experimental screening in a large ($\mathbf{q}, \omega$) region. This sets the band gap of liquid water at 9.3 eV. Upon performing a corresponding BSE calculation and accounting for finite size effects (see "Methods"), we find an exciton binding energy of 2.1 eV from the shift of the absorption onset, to be compared with the value of 1.64 eV calculated with the same definition in ref. 27. We calculate a macroscopic dielectric constant of 1.86, in good agreement with the experimental values of 1.76[8] and 1.78[17].

With the present estimate of the fundamental band gap, we also obtain other relevant energy levels of water. Following the scheme proposed in ref. 61, we position the electronic band structure of liquid water with respect to the vacuum level. We infer an ionization potential of 10.0 eV, in excellent accord with the reference value of 9.9 eV set by the photoemission experiments of Winter et al.[4] Our analysis situates the maximum of the $1b_1$ peak in the calculated density of states at 1.2 eV below the onset of the valence band, i.e., at 11.2 eV with respect to the

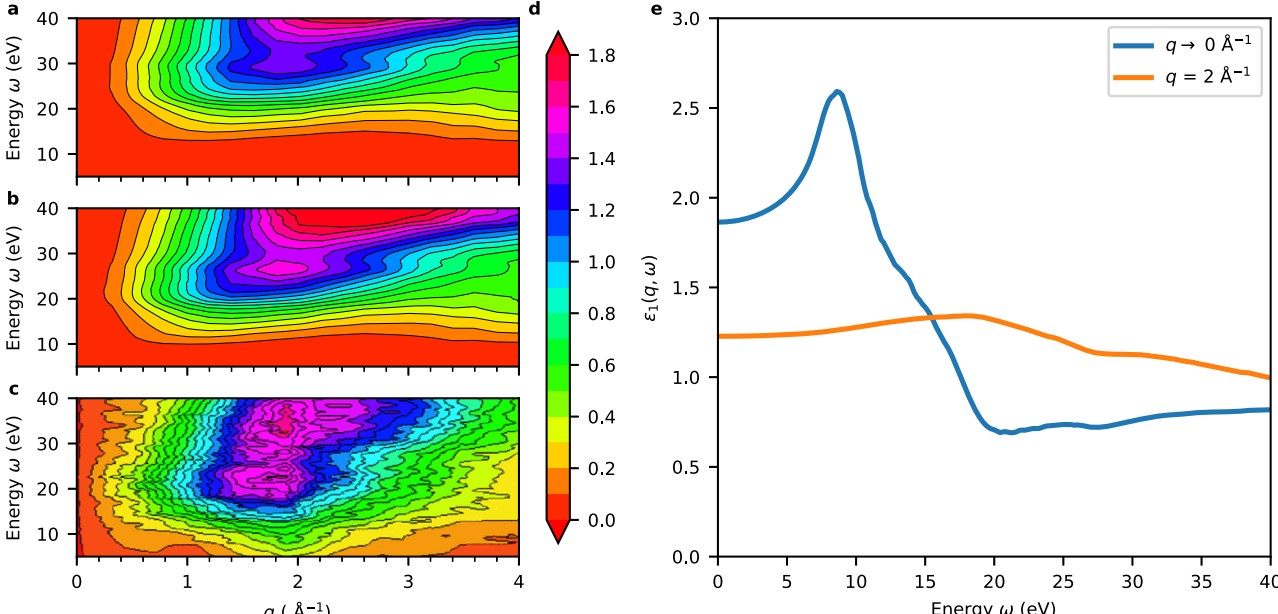

**Fig. 3 | Dielectric function of liquid water.** Contour plots of the opposite imaginary part of the susceptibility $-\mathrm{Im}[\chi(\omega, \mathbf{q})]$ in units of as Å$^{-3}$ obtained by solving the time-dependent density functional theory (TDDFT) equation (**a**) without any exchange-correlation $f_{xc}$ kernel and (**b**) with the $f_{xc}^*$ kernel, compared to (**c**) the experimental result from Abbamonte et al. (adapted with permission from ref. [16], Copyright (2004) by the American Physical Society); **d** Color scale pertaining to (**a**)–(**c**). **e** Real part of the dielectric function $\varepsilon_1(\mathbf{q}, \omega)$ for two values of transferred momentum as obtained from the TDDFT calculation with $f_{xc}^*$.

vacuum level, in very good agreement with the value of 11.16 measured by Winter et al.[4] but slightly lower than the measurements of 11.31 eV by Kurahashi et al.[62] and of 11.67 eV by Perrey et al.[63] The value of 0.7 eV for the electron affinity calculated in our work differs significantly from the recent theoretical estimate of $0.2 \pm 0.1$ eV put forward by Gaiduk et al.[7] The difference with respect to our result should be assigned to the larger band gap found in the calculations of Gaiduk et al.[7], in which no vertex corrections were included. From the experimental point of view, the electron affinity of liquid water has so far not been measured directly and has remained largely undetermined. A value of 0.97 eV could be extracted from thermodynamical data for the hydrated electron[64], quite distant from the values between 0.1 and 0.4 eV inferred from the average ejection length of the electrons in a two-photon ionization process[7]. The electron affinity calculated in this work situates in between these two estimates.

In conclusion, the dielectric function of liquid water is obtained for a wide range of transferred energies and momenta. The obtained function is validated through a comparison with absorption and IXS experiments, leading to a substantial improvement with respect to state-of-the-art methods. In addition, we demonstrate that the main resonance in the susceptibility does not correspond to a true plasmon, explaining the observed rapid decay of the excitation in real-time dynamics. The improved description of the electron screening then allows us to establish the fundamental band gap at 9.3 eV, the ionization potential at 10.0 eV, the electron affinity at 0.7 eV, and the exciton binding energy at 2.1 eV. Therefore, the present description of the screening allows one to cover a wide range of physical phenomena occurring in liquid water in an accurate and consistent manner.

## Table 1 | Band gaps of liquid water

|        | qs$GW_0$ | qs$GW$ | qs$G\bar{W}$ | qs$GW^*$ |
|--------|----------|--------|--------------|----------|
| $E_g$  | 8.7      | 9.8    | 9.1          | 9.3      |

The band gaps $E_g$ are calculated with qs$GW_0$ based on DFT screening, qs$GW$ with no vertex corrections, qs$G\bar{W}$ with bootstrap vertex corrections, and qs$GW^*$ obtained using the $f_{xc}^*$ kernel introduced in this work. Energies are given in eV.

## Methods

### Nuclear motion and electronic ground state

To account for the classical and quantum motion of oxygen and hydrogen atoms, we use 60 configurations evenly spaced in time from 6-bead path-integral molecular dynamics trajectories generated by Chen et al.[6,15] The configurations contain 32 water molecules. On top of them we first perform DFT calculations with the PBE exchange-correlation functional[65], which are necessary to achieve the $W_0$ screened potential. We use a plane-wave energy cutoff of 85 Ry and norm-conserving pseudopotentials[66]. Unless specified otherwise, the calculations are performed with the ABINIT code[67].

### GW calculations

For each configuration, we perform quasi-particle self-consistent $GW$ (qs$GW$) calculations, as first introduced by Faleev et al.[33] to correct the energies and wave functions obtained from DFT. This scheme takes advantage of the cancellation between the Z-factor and Γ in the self-energy[68]. In this framework, the main difference between $GW$ flavors reduces to the choice of $W$. We consider the non-self-consistent $W_0$ computed using PBE energies and wave functions, the self-consistent $W$ resulting from the iteration of energies and wave functions within qs$GW$, the vertex corrected $\bar{W}$, in which a bootstrap TDDFT kernel is additionally included[38], and $W^*$ computed using qs$GW$ self-consistent wave functions and energies and a vertex correction based on the long-range corrected (LRC) kernel[52–54] $f_{xc}^*$ defined in Eq. (4), where $\alpha = -10$ a.u. and $\beta = -0.2$ a.u. The values of $\alpha$ and $\beta$ are optimized to reproduce the experimental IXS spectrum at $q = 0.32$ Å$^{-1}$. When the values of $\alpha$ and $\beta$ are optimized to reproduce the imaginary part of the dielectric function, equivalent results are obtained (see Supplementary Discussion). We use an energy cutoff of 85 Ry for the exchange part of the self-energy and of 12 Ry for the correlation part. In the calculations of the dielectric function and of the self-energy, we include 2000 states, 550 of which are updated at each iteration. The screening is computed at 8 real and 4 imaginary frequencies. The $\mathbf{q} \to 0$ elements in the screening are calculated following the approach of Levine and Allan[69]. To obtain converged values, an extrapolation is carried out with respect to the number of updated states, the total

number of states, and the cutoff used in the correlation part of the self-energy (see Supplementary Methods). The band gap of liquid water corresponds to the difference between the conduction and valence band edges. The valence band edge is determined by linearly extrapolating the density of states[70], while the conduction band edge is taken as the energy level of the lowest unoccupied state following ref. [71], to overcome the finite-size effect due to the use of a supercell[70].

### BSE spectra

The BSE is solved in reciprocal space with the DP-EXC code[72]. The dielectric matrix is described by a $3400 \times 3400$ matrix in **G**-space and the wave functions are expanded on a basis of 28,257 plane waves. The Brillouin zone of the supercell is sampled at the $\Gamma$ point. A total of 128 occupied and 256 unoccupied states are included to achieve convergence of the optical spectra.

### TDDFT spectra

The TDDFT equation is solved in reciprocal space with the DP-EXC code[72]. A $2 \times 2 \times 2$ shifted Monkhorst-Pack grid is used to achieve a resolution of 0.3 Å$^{-1}$ in transferred momentum. To ensure convergence up to 40 eV, all the 128 occupied and 1300 unoccupied states are considered (total number of states of 1428). The dielectric matrix is described by a $1300 \times 1300$ matrix in **G**-space (cutoff of 6 Ry) and the wave functions are expanded on a basis of 28,257 plane waves (cutoff of 41 Ry). For the TDDFT spectra, convergence tests as a function of the number of bands, the energy cutoff of the wave functions, and the energy cutoff of the dielectric matrix are provided in Supplementary Methods.

### Determination of the exciton binding energy

To determine the exciton binding energy, we plot the absorption spectrum obtained using the BSE and the corresponding spectrum obtained in the RPA, see Supplementary Fig. 6. We then determine the exciton binding energy $E_b^{exc}$ from the separation between the onsets in the BSE and RPA spectra, following the same definition used in ref. [27]. For selected configurations, we checked that the qs$GW^*$ spectra close to the absorption onset only differ by a constant shift with respect to the corresponding qs$G\tilde{W}$ spectra, allowing us to extract the corresponding exciton binding energy. For qs$GW^*$, this gives $E_b^{exc} = 2.3$ eV. These values are to be corrected for finite size effects (see next section).

### Estimate of the finite-size correction to the exciton binding energy

To test the finite size effect on the exciton binding energy, we performed time-dependent PBE0 (TD-PBE0) calculations with a fraction of Fock exchange set to 0.40, in order to approximately reproduce the band gap of liquid water[73]. We compared the excitonic transition energy for the supercell in our work (cubic cell with a side of 9.84 Å, containing 32 water molecules) with the result for a $2 \times 2 \times 2$ supercell (cubic cell with a side of 19.68 Å, containing 256 water molecules). In constructing the latter system, we modified the structure of the extra replicas to ensure that they did not give rise to images of the localized exciton. Given the fact that the average exciton radius in ice has been found to be 4.0 Å[74], the result for the larger supercell cell can practically be considered as being converged. We find a finite-size correction of 0.21 eV leading to a reduced binding energy compared to that calculated with the small supercell. The size of this effect is fully consistent with the results of ref. [27], in which it was similarly found that the exciton binding energy was converged within 0.2–0.3 eV. The correction of 0.21 eV is used to correct the qs$GW^*$ exciton binding energy of 2.3 eV calculated with the small supercell. Accounting for this correction gives the estimate of 2.1 eV for the exciton binding energy. The TD-PBE0 calculations were performed with the CP2K suite of codes[75]. We used Goedecker-Teter-Hutter (GTH) pseudopotentials[76,77] and triple-zeta basis sets of MOLOPT quality[78]. The plane-wave energy cutoff for the electron density was set to 600 Ry to achieve converged energies. The Brillouin zones of the two supercells were sampled at the $\Gamma$ point. The hybrid functional calculations were performed with the auxiliary density matrix method to speed up the calculations[79].

## Data availability

Relevant data generated in this study have been deposited in the public repository Materials Cloud Archive, accessible at https://archive.materialscloud.org/record/2023.52.

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

## Acknowledgements

The authors thank Lucia Reining for useful discussions. Support from the Swiss National Science Foundation (SNSF) is acknowledged under Grant No. 200020-172524 (A.P.). This work was supported by grants from the Swiss National Supercomputing Centre (CSCS) with project IDs s879 and s1122 (A.P.).

## Author contributions

All authors contributed extensively to the realization of the work presented in this paper. I.R. and A.L. performed the calculations. A.P. supervised the work. All authors contributed to the writing of the manuscript.

## Competing interests

The authors declare no competing interests.
