## [Peer Review File · Nature Communications]

Many-body screening effects in liquid waterReviewers' comments:

Reviewer #1 (Remarks to the Author):

I have carefully read the manuscript "Many-body screening effects in liquid water: Implications for the band gap" by I. Reshetnyak and A. Pasquarello. The authors present an approach to simulate absorption and inelastic X-ray scattering spectra of liquid water. The main result reported in this paper is the description of a computational strategy that gives results in agreement with experimental results of liquid water (at specific temperature and pressure conditions), and by fitting experimental data.

The authors first show that they cannot assess which level of theory is optimal to describe liquid water's absorption spectrum because the latter is not so sensitive to the level of theory. The authors then turn to the discussion of inelastic X-ray scattering (IXS) of liquid water. They perform time dependent DFT calculations with an f_{xc} kernel that contains two parameters, which are determined by fitting the experimental IXS spectrum. It would be good if the authors could assign a physical meaning to the two parameters (alpha and beta). Do they represent physical quantities? About the fitting: the authors consider the experimental spectrum at a specific q ($q=0.32$). How did they choose q ? Can other values of q be chosen? If so, how are the results sensitive to this choice made by the authors? Is it possible to determine alpha and beta from first principles?

When the authors compute the ionization potential, they need to compute an electrostatic potential, which depends on the charge density. The authors use a quasi-particle self-consistent GW scheme to correct for deficiencies of DFT, which changes wave functions and, therefore, also updates the charge density. I would like to know if the authors have considered this effect when they computed the ionization potential, and if not, if they expect it to be negligible.

The authors imply that the electron affinity of water is 0.8 eV, which is distant from a recent theoretical estimate of 0.2 eV. Can this difference be rationalized? Is it due to size effects? The authors use 32 water molecules. Is this size enough? Perhaps the authors can try to verify if at the DFT level of theory (before applying post-DFT corrections) the results are similar, so as to attribute the difference solely to the different post-DFT treatment.

Do the authors agree with recent experimental measurements of the ionization energy of liquid water? Can they compute the vertical ionization energy of the b1 band and compare with : <https://doi.org/10.1021/acs.jpcllett.9b03391> ?

Can the presented approach be used to study other liquid systems?

As a final remark I urge the authors to better explain the implications of the results reported in the paper. In its current version the manuscript does not offer a substantial degree of novelty in terms of explanation or prediction of new effects in liquid water.

Reviewer #2 (Remarks to the Author):

This manuscript describes a first-principles investigation of the electronic screening effects in liquid water. Theoretical results are obtained by a variety of different approaches, including several flavors of TDDFT and BSE, and are compared to experimental absorption and inelastic X-ray scattering (IXS) spectra. The authors show that comparisons to the absorption spectra do not discriminate between the different theoretical approaches, whereas IXS provides a more useful benchmark. Based on this finding, the authors propose a new theoretical scheme that provides an improved theoretical description of IXS. This new approach is used to examine the electronic band gap of liquid water and is the basis for comparison between theory and experiment.

While the topic of the manuscript is interesting, I found the paper to be highly technical and in my view is unlikely to be of broad interest to the readers a journal like Nature Comm. A journal that specialize in the calculation of electronic properties of condensed matter, such as Phys. Rev or the

Journal of Physical Chemistry series would be more appropriate for this type of paper. Beyond this point, I have several specific comments and questions on the results and conclusions presented in the manuscript:

1. What is the underlying physics for the observation that optical spectra are less sensitive than IXS when comparing to different levels of theory? It is important to provide an explanation for this, rather than just presenting numerical results.

2. Related to the point (1). The authors should present their convergence check for the calculation of optical & IXS spectra. At the very least, this will help to eliminate the possibility that their conclusion concerning the sensitivity of IXS versus optical spectra on screening effects is simply an artifact of numerical convergence.

3. I'm not totally convinced that the introduction of a new f_{xc}^* truly provides a better agreement with experimental data for IXS. As shown in Fig. 2, while calculations with the revised f_{xc}^* (red) indeed improves the intensity of the calculated spectra over the original f_{xc} (orange), both of the results remain blue-shifted compared to the experimental one (green). In addition, the feature below 10 eV in the experimental spectrum is not reproduced by the new computational scheme.

4. The authors obtain a value of $E_b=2.3$ eV for the exciton binding energy of liquid water with their new qsGW* scheme. It is known that this value varies strongly with the size of the simulation cell (see Ref. 27 where the exciton binding energy was evaluated using system size of up to 256 water molecules). Accordingly, the results presented in this work for E_b are unlikely to be converged with respect to the simulation size.

5. The authors also present results for the band gap and electron affinity of liquid water using the new qsGW* scheme. These results, however, may not indicate that qsGW* is better than other theoretical approaches. For instance, an IP of 9.6 eV is smaller than the most recent experimental value of 9.9-10.0 eV, which is reproduced by the computational approach introduced in Ref. 7. In addition, the electron affinity obtained by qsGW* is -0.8 eV is quite different from a value of 0.1-0.3 eV below vacuum obtained in Ref. 7. It is also important to point out that an experimental value of the electron affinity of liquid water was revised in Ref. 7, where the latest value should be 0.1-0.4 eV below vacuum.

6. Related to (5), the authors will need to compare their results with the latest values for the electron affinity of water, much of which have been discussed in Ref. 7.

To conclude, the scope of the manuscript is not suitable to be considered in Nat. Comm. Besides the highly technical presentation of the results, I believe additional analysis is needed to confirm the convergence of the calculations and to justify the use of the new computational approach. Finally, the main conclusion concerning the electron affinity and band gap of liquid water needs further discussion and refinement.

Reviewer #3 (Remarks to the Author):

The manuscript titled "Many-body screening effects in liquid water: Implications for the band gap" by I. Reshentnyak and A. Pasquarello is a theoretical study aimed to determine accurate dielectric function of liquid water by means of many-body perturbation theory. The authors, by using state-of-the-art theoretical methods (Bethe Salpeter equation), first compare the absorption spectrum obtained using as starting point different flavours of quasiparticle self-consistent GW, concluding that such comparison it is not enough to assess the quality of the obtained dielectric function as the good agreement with the experimental spectrum obtained by qsGW and qpGW with a vertex correction is provided by compensation between many-body effects in the Green's function and in the screened electron-hole interaction. Next, a more stringent assessment is done by comparing the energy loss function and finally an ad-hoc TDDFT kernel obtained by optimising the loss

function with the experiment at a particular q point which is shown to provide a very good description of the dielectric function in a large range of momenta and energies, and a band-gap in agreement with the experiments.

The present work provides very interesting results benchmarking very accurate methodologies for a, particularly difficult case as the liquid water. Moreover, even if not in a completely ab-initio, an accurate description of the screening is provided that could serve as a benchmark for future works. For these reasons, the present manuscript may deserve the publication in Nature Communications, but before suggesting the publication I think that the authors should address the following points:

- 1) The loss function is calculated by means of TDDFT and three approximations are compared, anyway, also the loss function obtained by solving the Bethe-Salpeter equation using the two approximation shown in Fig.1 is accessible, in my opinion, in order to have a complete picture, also the performance of the BSE should be compared with the experiments and the TDDFT approximations.
- 2) The authors calculate a dielectric function that compares very well with IXS experiments by using an optimised kernel and next by using this results they do calculate electronic properties as band-gap and ionisation potential in agreement with the experiment. They also solve the BSE using such a screening finding a binding energy of 2.3 eV. Such binding energy would correspond to an excitation energy at 6.5eV at, which around 0.5eV lower than the onset of the experimental absorption. In my opinion, the author should present the obtained excitation spectra and comment on the agreement or disagreement with respect to the experimental absorption.
- 3) Figure 2: there is a bit of confusion between the terminology provided in the legend and the main text. While in the legend the spectra are indicated in terms of the used kernel, in the text they are referred with the used self-consistent GW energies and wave functions, as qsGW and $qsG\tilde{W}$. The two terminologies here are equivalent, as the vertex is built from the TDDFT equation reported in the text, but it would help the reader if the authors would state more explicitly which eigenvalues and eigenvectors are used to build X_{IP} in the various approximations.

Manuscript: NCOMMS-20-03699 (newly set to NCOMMS-20-03699A)

Title: Many-body screening effects in liquid water: Implications for the band gap

Authors: Igor Reshetnyak, Arnaud Lorin, and Alfredo Pasquarello

Detailed reply to the points raised by Reviewer #1:

1.1 It would be good if the authors could assign a physical meaning to the two parameters (alpha and beta). Do they represent physical quantities? Is it possible to determine alpha and beta from first principles?

The parameter α can be related to the inverse dielectric constant of the material. In particular, F. Sottile *et al.* Phys. Rev. B **68**, 205112, have shown that for an α/q^2 kernel to create weight within the quasi-particle band gap, α could be of the form: $\alpha = -4\pi / (\text{Re} [\epsilon(\omega_0)] - 1)$, where ω_0 is the frequency at which the exciton occurs. In the same work, the authors point out that the α/q^2 kernel leads to an overestimation of the spectral weight, and hence a violation of the sum rule. By introducing the β parameter, we obtain a re-scaling of the spectral weight by $(1+\beta)$: $\chi = \chi_1 + \chi_1^* \beta / \chi^* \chi = (1+\beta) \chi_1$, restoring the f sum rule. This gives a possible line for obtaining α and β from first principles.

We added a sentence in the manuscript to give a physical interpretation to α and β according to this reasoning.

1.2 The authors consider the experimental spectrum at a specific q ($q=0.32 \text{ \AA}^{-1}$). How did they choose q ? Can other values of q be chosen? If so, how are the results sensitive to this choice made by the authors?

More generally, one could optimize α and β by finding the best match over a wide range of q and ω . However, the f_{xc}^* derived from the data at $q = 0.32 \text{ \AA}^{-1}$ (left panel of Fig. 2) is found to lead to a comparable improvement of the description at $q = 0.95 \text{ \AA}^{-1}$ (right panel of Fig. 2). This indicates that the identified values of α and β lead to a consistent improvement of the agreement between the calculated and measured IXS spectrum over a wide range of q and ω . Such a global improvement suggests that the functional form of f_{xc}^* is able to describe the overall dependence of the loss function on q and ω .

In the revised manuscript, we now stress that the identified form of f_{xc}^* is able to describe the overall dependence of the loss function on q and ω .

1.3. When the authors compute the ionization potential, they need to compute an electrostatic potential, which depends on the charge density. The authors use a quasi-particle self-consistent GW scheme to correct for deficiencies of DFT, which changes wave functions and, therefore, also updates the charge density. I would like to know if the authors

have considered this effect when they computed the ionization potential, and if not, if they expect it to be negligible.

The changes in the electrostatic potential due to the difference between *GW* and DFT wave-functions can safely be neglected. For one of the most extreme cases, the Si-SiO₂ interface, Shaltaf *et al.* (Phys. Rev. Lett. **100**, 186401) have shown that the effect is of 20 meV, which is negligible. In our work, the ionization potential is determined through the scheme proposed in Ambrosio *et al.* (J. Phys. Chem. Lett. **9** 3212), which takes advantage of a PBE0(α) hybrid functional calculation.

1.4 The authors imply that the electron affinity of water is 0.8 eV, which is distant from a recent theoretical estimate of 0.2 eV. Can this difference be rationalized? Is it due to size effects?

Following a reconsideration of our results, the new value of the electron affinity has become 0.7 eV (see also our responses to points 1.6 and 2.5). The difference between our value and the recent theoretical estimate of 0.2 eV mainly results from the larger band gap obtained by Gaiduk *et al.* (Nature Commun. **9**, 247), who did not include vertex corrections in their calculations. We have effectively accounted for them in our work through the use of f_{xc} . It has been shown that the band edges of water can accurately be determined with the 32-molecules cell provided a linear extrapolation of the wings of the density of valence states is performed (J. Chem. Phys. **143**, 244508) (see also next point).

We added a sentence specifying that the origin of the difference between our value of the electron affinity and that found by Gaiduk *et al.* results from the treatment of vertex corrections.

1.5 The authors use 32 water molecules. Is this size enough? Perhaps the authors can try to verify if at the DFT level of theory (before applying post-DFT corrections) the results are similar, so as to attribute the difference solely to the different post-DFT treatment.

By comparing results for $N = 32, 64$ and 128 water molecules, Ambrosio *et al.* (J. Chem. Phys. **143**, 244508) showed that 32 water molecules are sufficient for obtaining accurate results for the conduction and valence band edges, provided that the valence band edge is identified by a proper extrapolation of the wings in the density of states. The conduction band edge is taken as the energy level of the lowest unoccupied state following Prendergast *et al.*, J. Chem. Phys. **123**, 014501 (2005). This is the approach we used in our work as far as the determination of the band gap is concerned. Hence, the results at the DFT level should be considered completely converged with cell size. Please refer to point 2.4, for a discussion on the convergence of the excitonic effects.

1.6 Do the authors agree with recent experimental measurements of the ionization energy of liquid water? Can they compute the vertical ionization energy of the b1 band and compare with : <https://doi.org/10.1021/acs.jpcllett.9b03391>?

Following this comment of the referee, we have carefully re-evaluated the extraction of the band gap using proper linear extrapolation of the wings in the density of states as suggested in Ambrosio *et al.* (J. Chem. Phys. **143**, 244508). This analysis has led to modified values for the band gap (9.3 eV with QSGW*), the ionization potential (10.0 eV), and the electron affinity (0.7 eV). In particular, from this analysis, we found that the $1b_1$ photoemission peak occurs at 1.2 eV from the onset of the valence band. This situates the $1b_1$ peak in our calculation at 11.2 eV, in good agreement with the measurement of 11.16 eV by Winter *et al.* [J. Phys. Chem A **108**, 2625 (2004)], but slightly lower than the measurements of 11.31 eV by Kurahashi *et al.* [J. Chem. Phys. **140**, 174506 (2014)] and of 11.67 eV by Perrey *et al.* [J. Phys. Chem. Lett. **11**, 1789 (2020)].

We added a sentence in the manuscript concerning the comparison with experiment for the $1b_1$ band. We have incorporated into the manuscript the new values for the band gap, the IP, and the EA, resulting from this analysis.

1.7 Can the presented approach be used to study other liquid systems?

There are no obstacles to extend the present approach to the study of other liquids. This should motivate more accurate measurements of their spectra.

1.8. As a final remark I urge the authors to better explain the implications of the results reported in the paper. In its current version the manuscript does not offer a substantial degree of novelty in terms of explanation or prediction of new effects in liquid water.

Our works aimed to improving the description of the many-body electron screening in a complex system such as liquid water. Our description could be validated through a comparison with the experimental IXS in an ample range of transferred energies and momenta, thereby demonstrating that the main resonance in the susceptibility does not correspond to a true plasmon and explaining the observed rapid decay of the excitation in real-time dynamics. This improved description of the electron screening then allows us to establish the fundamental band gap of liquid water at 9.3 eV, the ionization potential at 10.0 eV, the electron affinity at 0.7 eV and the exciton binding energy at 2.1 eV.

In the revised manuscript, we reformulated the conclusive paragraph along these lines to emphasize the novelty inherent to our results.

Detailed reply to the points raised by Reviewer #2:

2.1. What is the underlying physics for the observation that optical spectra are less sensitive than IXS when comparing to different levels of theory? It is important to provide an explanation for this, rather than just presenting numerical results.

In the computation of the optical spectra, the screened Coulomb interaction enters twice, but with opposite signs, i.e. in the GW quasi-particle corrections and in the

screened e-h interaction, leading to partial cancellation. In other terms, lower screening leads to an increased fundamental band gap, but concurrently increases the exciton binding energy. These changes have a tendency of compensating when focusing on the optical spectrum [Bechstedt et al., Phys. Rev. Lett. **78**, 1528 (1997)].

We added an explanatory sentence in the concerned paragraph.

2.2. Related to the point (1). The authors should present their convergence check for the calculation of optical & IXS spectra. At the very least, this will help to eliminate the possibility that their conclusion concerning the sensitivity of IXS versus optical spectra on screening effects is simply an artifact of numerical convergence.

The time-dependent density functional theory calculations of the optical and IXS spectra are converged with respect to the number of bands, the energy cutoff of the wave functions and the size of the dielectric matrix. It is therefore justified to consider the calculated spectra in the discussion. The higher sensitivity of the IXS spectrum stems from the extended energy range up to 40 eV over which it is considered, while the absorption spectrum is generally limited to the near-edge energy region.

In the revised manuscript, we now explicitly refer to the different energy ranges investigated in the absorption spectrum and in the IXS spectrum. This has led to a reformulation of the sentences in the abstract, and to an additional sentence when introducing the experimental IXS spectrum. We verified the convergence of the TDDFT spectra on the number of included bands, on the energy cutoff of the wave functions, and on the energy cutoff of the dielectric matrix. These convergence tests are mentioned in the method section and are provided in the Supplemental Information.

2.3. I'm not totally convinced that the introduction of a new f_{xc}^* truly provides a better agreement with experimental data for IXS. As shown in Fig. 2, while calculations with the revised f_{xc}^* (red) indeed improves the intensity of the calculated spectra over the original f_{xc} (orange), both of the results remain blue-shifted compared to the experimental one (green). In addition, the feature below 10 eV in the experimental spectrum is not reproduced by the new computational scheme.

Figure 1 shows that using no vertex correction (blue) or the Bootstrap f_{xc} (orange) both show reasonably good agreement with experiment. In Fig. 2, the energy positions of the peaks obtained with the Bootstrap f_{xc} (orange) noticeably improve the agreement with experiment compared with the case without f_{xc} (blue). This leads to the first conclusion that the inclusion of the vertex function improves the comparison with experiment. Furthermore, the revised f_{xc}^* (red) improves the agreement with experiment even more. Specifically, with the revised f_{xc}^* (red), the intensity of the loss function matches the experimental curve over a broad range of transferred energies and momenta, as shown in Fig. 2 for energies up to 40 eV and two different values of q .

Following the second query of the reviewer, we realized that the curve corresponding to the revised f_{xc}^* (red in Fig. 2) was not the correct one, even though still very similar

to the originally presented curve. Consideration of the proper curve also reproduces to some extent the intensity below 10 eV. However, it should be understood that it is necessary to describe the susceptibility over a large range of frequencies in order to properly capture the screening effects. In particular, the major contribution comes from the peak in the loss function, which occurs when the real part of the susceptibility is minimal or approaches zero [see Miglio et al., Eur. Phys. J. B **85**, 322 (2012)]. This generally occurs in correspondence of the plasmon frequency pole. In our case, the real part of the susceptibility is found not to vanish. Nevertheless, the critical energy region with the dominant contribution to the susceptibility is clearly identified to lie around 20 eV, as can be seen from Fig. 3. This implies that slight inaccuracies below 10 eV are not expected to contribute significantly to the screening effects.

We modified the text to clearly specify that the improved match with experiment for the intensity of the susceptibility is one of the key arguments which supports the use of the present f_{xc}^* kernel. We further remark that such an extended agreement around the intensity peak is important to properly capture the screening effects. We modified Figure 2 to include the correct curve corresponding to the case with the revised f_{xc}^* .

2.4. The authors obtain a value of $E_b=2.3$ eV for the exciton binding energy of liquid water with their new qsGW* scheme. It is known that this value varies strongly with the size of the simulation cell (see Ref. 27 where the exciton binding energy was evaluated using system size of up to 256 water molecules). Accordingly, the results presented in this work for E_b are unlikely to be converged with respect to the simulation size.

To test the finite size effect on the exciton binding energy, we performed TD-PBE0 calculations with a fraction of Fock exchange equal to 0.40, in order to approximately reproduce the band gap of liquid water [Ambrosio et al., J. Phys. Chem. B **120**, 7456 (2016)]. We compared the excitonic transition energy for the supercell in our work (cubic, $a_{lat}=9.84$ Å, 32 water molecules) with the result for a $2 \times 2 \times 2$ supercell (cubic, $a_{lat}=19.68$ Å, 256 water molecules). In constructing the latter system, we modified the structure of the extra replicas to ensure that they did not give rise to images of the localized exciton. Given the fact that the average exciton radius in ice has been found to be 4.0 angstrom [Hahn et al., Phys. Rev. Lett. **94**, 037404 (2005)], we expect that the result for the larger supercell cell is practically converged.

We find a difference of 0.21 eV, which leads to a smaller binding energy than calculated with the small supercell. The size of this effect is rather small and fully consistent with the results of Ref. 27, wherein the authors state : « Finally, we also performed calculations for a larger supercell including 256 water molecules (2048 valence electrons) and concluded that size effects, although not fully negligible, are rather minor on the value of the exciton binding energies (of the order of $\approx 0.2-0.3$ eV). » The correction that we calculated can be used to improve the estimate of the exciton binding energy found in our work.

In the revised manuscript, we included a sentence to specify that the estimate in our work for the exciton binding energy is 2.1 eV after consideration of the finite size

correction. In addition, we included in the Supplementary Information a small paragraph describing the extra calculation performed to estimate the finite size effect.

2.5 The authors also present results for the band gap and electron affinity of liquid water using the new qsGW* scheme. These results, however, may not indicate that qsGW* is better than other theoretical approaches. For instance, an IP of 9.6 eV is smaller than the most recent experimental value of 9.9-10.0 eV, which is reproduced by the computational approach introduced in Ref. 7. In addition, the electron affinity obtained by qsGW* is -0.8 eV is quite different from a value of 0.1-0.3 eV below vacuum obtained in Ref. 7. It is also important to point out that an experimental value of the electron affinity of liquid water was revised in Ref. 7, where the latest value should be 0.1-0.4 eV below vacuum.

This comment of Reviewer #2 spurred us to reconsider the extrapolations required for achieving the band edges from our spectra following more closely the analysis proposed in Ambrosio *et al.* (J. Phys. Chem. Lett. **9** 3212). This resulted in a revised value of 10.0 eV for the IP, in excellent agreement with the interval of 9.9-10.0 eV indicated by the Reviewer. We added a sentence to the manuscript to highlight this agreement.

The new analysis results in a electron affinity of 0.7 eV, still significantly larger than in Ref. 7. This difference should be assigned to the larger band gap found in Ref. 7, in which no vertex corrections were considered (see also our response to point 1.4).

From the experimental point of view, the electron affinity of liquid water has so far not been measured directly and has remained largely undetermined. A value of 0.97 eV could be extracted from thermodynamical data for the hydrated electron (Ambrosio *et al.*, J. Phys. Chem. Lett. **8**, 2055), quite distant from the value between 0.1 and 0.4 eV inferred from the average ejection length of the electrons in a two-photon ionization process (Gaiduk *et al.*, Nature Commun. **9**, 247). The electron affinity calculated in this work situates in between these two estimates.

In the revised version of our manuscript, we reformulate the paragraph in which the electron affinity is discussed, pointing out various values extracted from experiment.

2.6. Related to (5), the authors will need to compare their results with the latest values for the electron affinity of water, much of which have been discussed in Ref. 7.

As described in our response to point 2.5, we have added a discussion to compare our results for the electron affinity with the theoretical results of Gaiduk *et al.*, Nature Commun. **9**, 247 (Ref. 7) and of Ambrosio *et al.*, J. Phys. Chem. Lett. **9**, 3212 (2018), together with various values extracted from the experiment.

Detailed reply to the points raised by Reviewer #3:

3.1 The loss function is calculated by means of TDDFT and three approximations are compared, anyway, also the loss function obtained by solving the Bethe-Salpeter equation

using the two approximations shown in Fig.1 is accessible, in my opinion, in order to have a complete picture, also the performance of the BSE should be compared with the experiments and the TDDFT approximations.

Because of its computational cost, BSE cannot be performed throughout the large energy range required to properly assess the loss function. However, to address the request of the Reviewer for a global comparison, we compare the absorption spectra obtained with BSE and TDDFT over the frequency range in Fig. 1. Being Fig. 1 already very crowded, this comparison is given in Fig. S3 of the Supplementary Information. It clearly appears that the TDDFT spectrum does not accurately reproduce the detailed shape of the experimental spectrum near the onset, lacking the excitonic contributions. However, the TDDFT calculation is sufficiently accurate to provide a good description of the loss function $\text{Im}[\epsilon^{-1}]$ over an extended range of frequencies, as can be seen in Fig. 2 of the main text.

3.2 The authors calculate a dielectric function that compares very well with IXS experiments by using an optimised kernel and next by using these results they do calculate electronic properties as band-gap and ionisation potential in agreement with the experiment. They also solve the BSE using such a screening finding a binding energy of 2.3 eV. Such binding energy would correspond to an excitation energy at 6.5 eV at, which is around 0.5 eV lower than the onset of the experimental absorption. In my opinion, the author should present the obtained excitation spectra and comment on the agreement or disagreement with respect to the experimental absorption.

In the case of a disordered system like liquid water, the excitonic contributions to the spectrum do not result in a narrow exciton peak as in the case of semiconductors, but rather result in a broad peak with a range of exciton binding energies. Thus, the exciton binding energy obtained in our work follows the definition introduced previously by Nguyen *et al.* in Phys. Rev Lett. **122**, 237402 (2019) for liquid water. The exciton binding energy is defined as the difference between the onsets of the BSE and RPA spectra. In this way, we find an exciton binding energy of 2.3 eV (before corrections due to finite size) when using the qsGW* scheme, to be compared with the result of 1.64 eV obtained by Nguyen *et al.* in Phys. Rev Lett. **122**, 237402 (2019). To further clarify this aspect, we have graphically illustrated the definition of exciton binding energy used in this work in Fig. S2 of the Supplementary Information.

The BSE calculation includes the excitonic contributions from the outset. Hence, the quality of the agreement between theory and experiment can directly be based on the comparison between the BSE and the experimental curves in Fig. 1. It is already commented in the manuscript that « good agreement between theory and experiment is achieved when either qsGW or qsGW~ are used as starting points in the BSE ».

3.3 Figure 2: there is a bit of confusion between the terminology provided in the legend and the main text. While in the legend the spectra are indicated in terms of the used kernel, in the text they are referred with the used self-consistent GW energies and wave functions, as qsGW and qsG\tilde{W}. The two terminologies here are equivalent, as the vertex is built

from the TDDFT equation reported in the text, but it would help the reader if the authors would state more explicitly which eigenvalues and eigenvectors are used to build X_{IP} in the various approximations.

We thank the referee for pointing this out. We have modified the text in the revised manuscript to present the results in a coherent way with respect to the legend in Figure 2.

REVIEWER COMMENTS

Reviewer #2 (Remarks to the Author):

I have read through the revised manuscript and the responses to the questions that I had asked earlier. My overall assessment is that the quality of the paper has been improved and I believe all of the technical questions that I had have been adequately addressed. My original concern that the paper is perhaps too technical to be of broad interest to the readers of a journal like Nature still remains, but I want to stress that this is a minor point.

Reviewer #3 (Remarks to the Author):

I've read the revised version of the manuscript and in my opinion the authors have answered satisfactorily to all the points raised by all the reviewers. I can suggest publication in Nature Communication, even if I would like to insist that a BSE calculation of the IXS spectrum would be very useful. The authors claim that such calculations are too expensive, but I believe that this kind of calculation is feasible, e.g. recurring to codes that solve the BSE with iterative algorithms.

Alternatively, how the IXS spectra would compare with experiments in case α and β are determined through optimising the dielectric function (absorption).

Finally, in the SI convergence studies reported, for completeness I also suggest adding the convergence study done for the GW calculation showing the extrapolations done with respect to the number of updated states, the total number of states, and the cut-off used in the correlation part of the self-energy.

Manuscript: NCOMMS-20-03699A

Title: Many-body screening effects in liquid water: Implications for the band gap

Authors: Igor Reshetnyak, Arnaud Lorin, and Alfredo Pasquarello

Lausanne, 7 February 2023

Detailed response to the points raised by Reviewer #3:

3.1 I can suggest publication in Nature Communication, even if I would like to insist that a BSE calculation of the IXS spectrum would very useful. Authors claims that such calculations are too expensive, but I believe that this kind of calculation is feasible, e.g. recurring to codes that solves the BSE with iterative algorithms. Alternatively, how the IXS spectra would compare with experiments in case α and β are determined through optimising the dielectric function(absorption).

Calculating the full dielectric function $\epsilon(\mathbf{q},\omega)$ averaged over a sufficient number of configurations of liquid water remains a formidable computational task, even with codes using the most modern iterative algorithms. Indeed, for achieving the \mathbf{q} dependence, it is necessary to use a finer \mathbf{k} -point grid than just the Γ point, notably at least $2\times 2\times 2$ to achieve a resolution of 0.3 \AA^{-1} in the transferred momentum. Furthermore, it is necessary to achieve a two-dimensional description in the (\mathbf{q},ω) space with large sampled intervals in both \mathbf{q} and ω for obtaining the result provided in Fig. 3. Finally the calculations, should be repeated for several configurations of liquid water to account for the thermal disorder. The strength of our work consists in showing that it is possible to turn from the four-point dielectric susceptibility $^4\chi$ to the two-point susceptibility $^2\chi$, overcoming in this way the necessity of lengthy calculations.

Nevertheless, we now performed extra calculations to extend the range of frequencies of the BSE-qsGW spectra of the imaginary part of the dielectric function ϵ_2 up to 20 eV, covering in this way the relevant region of the principal peak and allowing for a wider comparison with the TDDFT spectrum in Fig. S7. While it clearly appears that the TDDFT spectrum does not reproduce the detailed shape of the experimental spectrum near the onset due to the lack of excitonic contributions, the TDDFT calculation nonetheless is able to reproduce the position of the main peak. Furthermore, as seen in Fig. 2 of the main text, the TDDFT calculation is sufficiently accurate to provide a good description of the loss function $\text{Im}[\epsilon^{-1}]$ over an extended range of frequencies.

In addition, we pursue the alternative suggestion put forward by Reviewer #3, which consists in determining the parameters α and β through optimising the imaginary part of the dielectric function ϵ_2 . First, we show that the adopted choice for α and β already gives a reasonably accurate description of ϵ_2 . (cf. Fig. S3). Next, for a representative configuration of liquid water, we study how ϵ_2 can be further improved by optimising α and β (cf. Figs. S3 and S4). Finally, we check that the

values of α and β obtained in this way still lead to an accurate description of the IXS spectrum at finite values of transferred momentum \mathbf{q} (cf. Fig. S5). These results indicate that the dielectric function is overall well described with a specific choice of α and β and that it remains robust upon small variations of these parameters.

To address this point, we brought the following changes to the manuscript. First, we extended the range of frequencies of the BSE-qsGW spectra of ϵ_2 up to 20 eV in Fig. S7. Next, we revised the main text to explicitly invoke the necessity of describing a large range of transferred momenta for justifying the fact of turning to $^2\chi$ when investigating the full dielectric function. Furthermore, we included the effect of varying α and β upon the absorption spectrum in the Supplementary Information (Figs. S3-S5). In particular, we give the analogous of Fig. 2, i.e. the loss function for the values of α and β , which result from the optimisation of ϵ_2 (Fig. S5), as suggested by the Reviewer.

3.2 Finally, in the SI convergence studies are reported, for completeness I also suggest adding the convergence study done for the GW calculation showing the extrapolations done with respect to the number of updated states, the total number of states, and the cut-off used in the correlation part of the self-energy.

For completeness, the three convergence studies associated with the GW calculations have been added to the Supplementary Information, as suggested.

REVIEWERS' COMMENTS

Reviewer #3 (Remarks to the Author):

In my opinion, the authors have responded satisfactorily to all the points raised. For this reason, I suggest the publication of the manuscript in Nature Communication

Manuscript: NCOMMS-20-03699B

Title: Many-body screening effects in liquid water

Authors: Igor Reshetnyak, Arnaud Lorin, and Alfredo Pasquarello

Lausanne, 28 March 2023

Response to comments of Reviewer #3:

In my opinion, the authors have responded satisfactorily to all the points raised. For this reason, I suggest the publication of the manuscript in Nature Communication

We thank the Reviewer for his careful comment.